# Molecular Phylogeny and Morphology Reveal Cryptic Species in the *Cordyceps militaris* Complex from Vietnam

**DOI:** 10.3390/jof9060676

**Published:** 2023-06-15

**Authors:** Yao Wang, Quan-Ying Dong, Run Luo, Qi Fan, Dong-E Duan, Van-Minh Dao, Yuan-Bing Wang, Hong Yu

**Affiliations:** 1Yunnan Herbal Laboratory, College of Ecology and Environmental Sciences, Yunnan University, Kunming 650504, China; wangyao1@aliyun.com (Y.W.); nympheel@163.com (Q.-Y.D.); luorun0214@163.com (R.L.); 2The International Joint Research Center for Sustainable Utilization of Cordyceps Bioresources in China and Southeast Asia, Yunnan University, Kunming 650504, China; 15925137267@139.com (Q.F.); wangyb001@126.com (Y.-B.W.); 3Institute of Regional Research and Development, Ministry of Science and Technology, Hanoi 100803, Vietnam; minhdaovan87@gmail.com

**Keywords:** new taxa, macro-morphology, phylogenetic analyses, arthropod-pathogenic fungi

## Abstract

The *Cordyceps militaris* complex, which is a special group in the genus *Cordyceps*, is rich in species diversity and is widely distributed in nature. Throughout the investigations of arthropod-pathogenic fungi in the national reserves and in Vietnam parks, collections of *C. militaris* attacking lepidopteran pupae or larvae were located in the soil and on the leaf litter. The phylogenetic analyses of the combined nr*SSU*, nr*LSU*, *TEF*, *RPB1*, and *RPB2* sequence data indicated that the fungal materials collected in Vietnam belonged to *C. militaris* and two hidden species in the *C. militaris* complex. The phylogenetic analyses and morphological comparisons presented here strongly supported the descriptions of *C. polystromata* and *C. sapaensis* as new taxa as well as *C. militaris* as a known species. The morphological characteristics of 11 species in the *C. militaris* complex, which included two novel species and nine known taxa, were also compared.

## 1. Introduction

The genus *Cordyceps* is a very important fungal resource, with some of the species having economic, medicinal, and ecological importance. *Cordyceps chanhua* has been applied as a tonic and medicinal food in traditional Chinese medicine (TCM) for a long time due to its significant health benefits [1]. “Mister Lei’s Treatise on Processing Drugs” recorded *C. chanhua* as a TCM nearly 1500 years ago (Chinese Southern and Northern Dynasties). *Cordyceps tenuipes* are widely used as a raw material in functional foods in Japan and South Korea, because of their important immune-regulatory, antitumor, analgesic, antibacterial, antimalaria, and other pharmacological effects [2]. *Cordyceps fumosorosea* are an important environmentally safe alternative to chemical pesticides for the interspecific transmission and biological control of pest insects [3,4]. One of the best-known edible and medicinal cordycipitoid fungi is *C. militaris*; it has been used in East Asian countries, including China, Korea, and Japan, for many years and it is currently widely used in Western countries [5,6]. In recent years, research has shown that *C. militaris* possesses different biologically active ingredients that are beneficial to the human body; these include cordycepin, cordycepic acid, cordyceps polysaccharides, pentastatin, and carotenoids [7,8,9]. Currently, a vast *C. militaris* industry exists in China that has created an estimated economic value of 10 billion Chinese Yuan annually [6].

*Cordyceps militaris* is the oldest accepted scientific name, although this species was described in the 17th- and early 18th-century literature under the old and now obsolete generic names *Fungus* and *Fungoides* [10]. The species was transferred to *Clavaria* by Linnaeus (1753) and then again transferred to the ascomycete genus *Sphaeria* by Ehrhart (1791) (https://www.mycobank.org, accessed on 20 April 2023), a classification that was followed up until the early 19th century [10]. In 1818, this species was then transferred to its own genus, and *Cordyceps* Fries was erected on the basis of *C. militaris* as the type species [11].

*Cordyceps sensu lato* (*s. l.*) is an important worldwide group of arthropod-pathogenic fungi, with the height of known species diversity in subtropical and tropical regions [12,13]. Vietnam is located in a tropical region that has an extremely rich biodiversity in Southeast Asia. These forests feature a rich biodiversity of flora and fauna because of the tropical monsoon climate with high temperatures and rainfall [14]. These conditions have produced a favorable environment for the development of arthropod-pathogenic fungi. Currently, over 100 species of *Cordyceps s. l.* have been reported, but this is only a small portion of arthropod-pathogenic fungi resources in Vietnam [15]. Throughout the investigations of arthropod-pathogenic fungi in national reserves and Vietnam parks, collections of *C. militaris* attacking lepidopteran pupae or larvae were located in the soil or on the leaf litter. The collections have various phenotypic characters, including numbers and lengths of stromata, and shapes and colors of fertile parts. Consequently, in the present study, it was hypothesized that *C. militaris* from Vietnam is a species complex. This study aims to clarify the hidden species in *C. militaris* via molecular phylogenetic studies and to combine molecular analyses with observations of diagnostic morphological characters.

## 2. Materials and Methods

### 2.1. Specimen Collection and Fungus Isolation

Fungus-infected insect specimens were collected in two locations in Sa Pa District, Lao Cai Province, Vietnam. The specimens were collected by carefully unearthing their hosts with a scoop and then placing the samples in sterile bags. The collection site information was documented, including the altitude, longitude, latitude, and habitat type. Teleomorph specimens were rinsed with tap water, washed with sterile distilled water, and then dried on sterile filter paper. A mass of ascospores and asci was removed from the perithecia with a fine needle and placed in a drop of sterile water, which was then stirred with a needle to evenly distribute the elements on the slide. To acquire monospore cultures, a portion of the drop containing ascospores was placed on plates of potato dextrose agar (PDA; potato 200 g/L, dextrose 20 g/L, agar 20 g/L) using a sterile micropipette, and then the Petri dish was incubated at 25 °C. The purified fungal strains were maintained in a culture room at 25 °C or were transferred to PDA slants and stored at 4 °C. Voucher specimens and the corresponding isolated strains were deposited in the Yunnan Herbal Herbarium (YHH) and the Yunnan Fungal Culture Collection (YFCC), respectively, of Yunnan University, Kunming, China.

### 2.2. Morphological Observations

Macro-morphological characteristics, including the host, fungi location, color and shape of the stromata, and perithecial orientation (superficial, immersed, or semi-immersed; ordinal or oblique), were examined under a dissecting microscope (SZ61, Olympus Corporation, Tokyo, Japan). Cultures on slants were transferred to PDA plates and cultured in an incubator for 14 days at 25 °C. The cultures were observed for the comparison of important morphological characters, including the conidial arrangement, phialides, and colony pigmentation. For morphological evaluation, microscope slides were prepared by placing mycelia from the cultures on PDA blocks (5 mm diameter) and then they were overlaid with a coverslip. Medan dye solution was used to observe asci and ascospores. Other structures were mounted in water. The sizes and shapes of the asexual morphological characteristics, including the conidiophores, phialides, and conidia, were determined using a light microscope (CX40, Olympus Corporation, Tokyo, Japan) and a scanning electron microscope (Quanta 200 FEG, FEI Company, Hillsboro, OR, USA).

### 2.3. DNA Extraction, Polymerase Chain Reaction (PCR), and Sequencing

Specimens and live axenic cultures were prepared for DNA extraction to ensure that both represented the same species. Genomic DNA was extracted with a Genomic DNA Purification Kit (Qiagen GmbH, Hilden, Germany) based on the manufacturer’s protocol. The primer pair nrSSU-CoF and nrSSU-CoR [16] was used to amplify the nuclear ribosomal small subunit (nr*SSU*); the primer pair LR5 and LR0R [17,18] was used to amplify the nuclear ribosomal large subunit (nr*LSU*); and the primer pair EF1α-EF and EF1α-ER [12,19] was used to amplify the translation elongation factor 1α (*TEF*). For amplification of the largest and second-largest subunits of RNA polymerase II (*RPB1* and *RPB2*), the PCR primer pairs RPB1-5′F/RPB1-5′R and RPB2-5′F/RPB2-5′R [12,19] were employed. The primers used for PCR amplification of nr*SSU*, nr*LSU*, *TEF*, *RPB1*, and *RPB2* are listed in Appendix A. All of the PCR reactions were performed in a final volume of 50 μL containing 25 μL 2 × Taq PCR Master Mix (Tiangen Biotech Co., Ltd., Beijing, China), 0.5 μL of every primer (10 μM), 1 μL of genomic DNA, and 23 μL of RNase-Free water. The PCR products were sequenced by Beijing Sinogenomax Co., Ltd., Beijing, China.

### 2.4. Phylogenetic Analyses

Phylogenetic analyses were based on the sequences of five loci (nr*SSU*, nr*LSU*, *TEF*, *RPB1*, and *RPB2*). All of the sequences, which were retrieved from GenBank, were combined with those generated in the study. Sequences were aligned with MAFFT v.7 (http://mafft.cbrc.jp/alignment/server/ (accessed on 20 April 2023)). The aligned sequences were then manually corrected where necessary. Following sequence alignment, the aligned sequences of five genes were concatenated. A partition homogeneity test was performed with PAUP * 4.0a166 [20], and the result revealed that there was no significant conflict among the different data partitions. The PartitionFinder V2.0.0 program was used to identify five data partitions, including one for both nr*SSU* and nr*LSU*, and four for each of the three codon positions for the protein-coding genes *TEF*, *RPB1*, and *RPB2* [21]: Partition 1—nr*SSU* and nr*LSU*; Partition 2—*TEF*_pos1 and *TEF*_pos2; Partition 3—*RPB1*_pos1 and *RPB2*_pos1; Partition 4—*RPB1*_pos2 and *RPB2*_pos2; and Partition 5—*TEF*_pos3, *RPB1*_pos3, and *RPB2*_pos3. The best substitution models of these five partitions were calculated with jModelTest version 2.1.4 [22]. The GTR + G + I model was used for the partitions of nr*SSU*, nr*LSU*, *TEF*_pos1, *TEF*_pos2, *TEF*_pos3, *RPB1*_pos3, and *RPB2*_pos3, and the GTR + I model was used for partitions of *RPB1*_pos1, *RPB1*_pos2, *RPB2*_pos1, and *RPB2*_pos2. Maximum likelihood (ML) phylogenetic analyses were conducted in RaxML 7.0.3 [23] with the recommended partition parameters, and 1000 rapid bootstrap replicates were performed on the dataset. Bayesian inference (BI) analysis was conducted using MrBayes v3.1.2 for five million generations with the GTR + G + I model, and the model was employed separately for each of the five gene partitions [24].

## 3. Results

### 3.1. Sequencing and Phylogenetic Analyses

The analyzed data matrix employed to construct the phylogeny of *Cordyceps* species included sequences from 102 samples (Table 1). *Liangia sinensis* YFCC 3103 and *L. sinensis* YFCC 3104 were utilized as outgroups. The final dataset consisted of 4627 bp of sequence data, including gaps (nr*SSU* 1060 bp, nr*LSU* 877 bp, *TEF* 999 bp, *RPB1* 719 bp, and *RPB*2 972 bp). Both the BI and ML analyses produced trees with similar topologies that resolved the majority of the *Cordyceps* lineages in separate terminal branches (Figure 1). The phylogenetic trees indicated that these were identical in overall topologies with the findings of prior work [25,26] and revealed the species diversity of the *C. militaris* complex in *Cordyceps* clades. The analyses also revealed that two newly discovered species, *C. polystromata* and *C. sapaensis*, were phylogenetically clustered with *C. chaetoclavata* (H. Yu et al.), *C. inthanonensis* (Mongkols. et al.), and *C. rosea* (Kobayasi and Shimizu), but they were distinguished from the latter three by forming two separate branches in the *C. militaris* complex. These results indicated that the *C. militaris* complex should be composed of 11 species, namely, *C. chaetoclavata*, *C. inthanonensis*, *C. kyusyuensis* Kawam., *C. militaris*, *C. ningxiaensis* (T. Bau and J.Q. Yan), *C. oncoperae* (P.J. Wright), *C. polystromata*, *C. rosea*, *C. roseostromata* (Kobayasi and Shimizu), *C. sapaensis*, and *C. shuifuensis* H. Yu et al.

**Table 1 jof-09-00676-t001:** Specimen information and GenBank accession numbers for sequences used in this study.

Species List	Voucher Information	Host/Substrate	GenBank Accession Number					Reference
			nr*SSU*	nr*LSU*	*TEF*	*RPB1*	*RPB2*	
*Cordyceps albocitrinus*	spat 07-174		MF416575		MF416467	MF416629		[27]
*Cordyceps amoene-rosea*	CBS 107.73	Coleoptera	AY526464	MF416550	MF416494	MF416651	MF416445	[27,28]
*Cordyceps amoene-rosea*	CBS 729.73	Coleoptera	MF416604	MF416551	MF416495	MF416652	MF416446	[27,28]
*Cordyceps araneae*	BCC 85065	Arachnid		MT003037	MT017850	MT017810	MT017828	[29]
*Cordyceps araneae*	BCC 85066	Arachnid		MT003038	MT017851	MT017811	MT017829	[29]
*Cordyceps araneae*	BCC 88291	Arachnid		MT003039	MT017852	MT017812	MT017830	[29]
*Cordyceps bifusispora*	spat 08-129	Lepidoptera	MF416576	MF416523	MF416468	MF416630		[27]
*Cordyceps bifusispora*	spat 08-133.1	Lepidoptera	MF416577	MF416524	MF416469	MF416631	MF416434	[27]
*Cordyceps bifusispora*	EFCC 5690	Lepidoptera	EF468952	EF468806	EF468746	EF468854	EF468909	[12]
*Cordyceps bifusispora*	EFCC 8260	Lepidoptera	EF468953	EF468807	EF468747	EF468855	EF468910	[12]
*Cordyceps blackwelliae*	TBRC 7255	Lepidoptera		MF140703	MF140823	MF140772	MF140796	[30]
*Cordyceps blackwelliae*	TBRC 7256	Coleoptera		MF140702	MF140822	MF140771	MF140795	[30]
*Cordyceps blackwelliae*	YFCC 856	Lepidoptera	MW181780	MW173992	MW168233	MW168199	MW168216	[31]
*Cordyceps brevistroma*	BCC 78209	Lepidoptera		MT003044	MT017855	MT017817	MT017835	[29]
*Cordyceps brevistroma*	BCC 79253	Lepidoptera		MT003045	MT017856	–	MT017836	[29]
*Cordyceps buttonspora*	YFCC 8400	Lepidoptera	OL468555	OL468575	OL473523	OL739569	OL473534	[25]
*Cordyceps buttonspora*	YFCC 8401	Lepidoptera	OL468556	OL468576	OL473524	OL739570	OL473535	[25]
*Cordyceps caloceroides*	MCA 2249	Araneae	MF416578	MF416525	MF416470	MF416632		[27]
*Cordyceps cateniannulata*	CBS 152.83	Coleoptera	AY526465	MG665226	JQ425687			[30,32]
*Cordyceps cateniobliqua*	YFCC 3367	Coleoptera	MN576765	MN576821	MN576991	MN576881	MN576935	[26]
*Cordyceps cateniobliqua*	YFCC 5935	Lepidoptera	MN576766	MN576822	MN576992	MN576882	MN576936	[26]
*Cordyceps cateniobliqua*	CBS 153.83	Lepidoptera	AY526466		JQ425688		MG665236	[30,32]
*Cordyceps* cf. *ochraceostromata*	ARSEF 5691	Lepidoptera	EF468964	EF468819	EF468759	EF468867	EF468921	[12]
*Cordyceps* cf. *pruinosa*	spat 08-115	Lepidoptera	MF416586	MF416532	MF416476	MF416635	MF416439	[27]
*Cordyceps* cf. *pruinosa*	spat 09-021	Lepidoptera	MF416587	MF416533	MF416477	MF416636		[27]
*Cordyceps* cf. *pruinosa*	NHJ 10627	Lepidoptera	EF468967	EF468822	EF468763	EF468870		[12]
*Cordyceps* cf. *pruinosa*	NHJ 10684	Lepidoptera	EF468968	EF468823	EF468761	EF468871		[12]
*Cordyceps* cf. *pruinosa*	EFCC 5693	Lepidoptera	EF468966	EF468821	EF468762	EF468869		[12]
*Cordyceps* cf. *pruinosa*	EFCC 5197	Lepidoptera	EF468965	EF468820	EF468760	EF468868		[12]
*Cordyceps* cf. *takaomontana*	NHJ 12623	Lepidoptera	EF468984	EF468838	EF468778	EF468884	EF468932	[12]
*Cordyceps chaetoclavata*	YHH 15101	Lepidoptera	MN576722	MN576778	MN576948	MN576838	MN576894	[26]
*Cordyceps chiangdaoensis*	BCC 68469	Coleoptera		MF140732	KT261403			[30,33]
*Cordyceps chiangdaoensis*	YFCC 857	Coleoptera	MW181781	MW173993	MW168234	MW168200	MW168217	[31]
*Cordyceps cicadae*	GACP 07071701	Hemiptera	MK761207	MK761212	MK770631			[34]
*Cordyceps cicadae*	RCEF HP090724-31	Hemiptera	MF416605	MF416552	MF416496	MF416653	MF416447	[27]
*Cordyceps cocoonihabita*	YFCC 3415	Lepidoptera	MN576723	MN576779	MN576949	MN576839	MN576895	[26]
*Cordyceps cocoonihabita*	YFCC 3416	Lepidoptera	MN576724	MN576780	MN576950	MN576840	MN576896	[26]
*Cordyceps coleopterorum*	CBS 110.73	Coleoptera	JF415965	JF415988	JF416028	JN049903	JF416006	[35]
*Cordyceps exasperata*	MCA 2288	Lepidoptera	MF416592	MF416538	MF416482	MF416639		[27]
*Cordyceps farinosa*	CBS 111113	–	AY526474	MF416554	MF416499	MF416656	MF416450	[27,32]
*Cordyceps fumosorosea*	YFCC 4561	Lepidoptera	MN576761	MN576817	MN576987	MN576877	MN576931	[26]
*Cordyceps fumosorosea*	CBS 244.31	Butter	MF416609	MF416557	MF416503	MF416660	MF416454	[27]
*Cordyceps fumosorosea*	CBS 375.70	Food			MF416501	MF416658	MF416452	[27]
*Cordyceps fumosorosea*	CBS 107.10	–		MG665227	HM161735		MG665237	[28]
*Cordyceps grylli*	MFLU 17-1023	Orthoptera	MK863048	MK863055	MK860193			Unpublished
*Cordyceps grylli*	MFLU 17-1024	Orthoptera	MK863049	MK863056	MK860194			Unpublished
*Cordyceps inthanonensis*	BCC 79828	Lepidoptera		–	MT017854	MT017816	MT017833	[29]
*Cordyceps inthanonensis*	BCC 56302	Lepidoptera		MT003040	MT017853	MT017814	MT017831	[29]
*Cordyceps inthanonensis*	BCC 55812	Lepidoptera		MT003041	–	MT017815	MT017832	[29]
*Cordyceps jakajanicola*	BCC 79816	Hemiptera		MN275696	MN338479	MN338484	MN338489	[36]
*Cordyceps jakajanicola*	BCC 79817	Hemiptera		MN275697	MN338480	MN338485	MN338490	[36]
*Cordyceps javanica*	TBRC 7259	Lepidoptera		MF140711	MF140831	MF140780	MF140804	[30]
*Cordyceps javanica*	CBS 134.22	Coleoptera	MF416610	MF416558	MF416504	MF416661	MF416455	[27]
*Cordyceps kuiburiensis*	BCC 90322	Araneidae		MK968816	MK988032	MK988030		[36]
*Cordyceps kuiburiensis*	BCC 90323	Araneidae		MK968817	MK988033	MK988031		[36]
*Cordyceps kyusyuensis*	EFCC 5886	Lepidoptera	EF468960	EF468813	EF468754	EF468863	EF468917	[12]
*Cordyceps lepidopterorum*	TBRC 7263	Lepidoptera		MF140699	MF140819	MF140768	MF140792	[30]
*Cordyceps lepidopterorum*	TBRC 7264	Lepidoptera		MF140700	MF140820	MF140769	MF140793	[30]
*Cordyceps longiphialide*	YFCC 8402	rotted wood	OL468557	OL468577	OL473525	OL739571	OL473536	[25]
*Cordyceps longiphialide*	YFCC 8403	rotted wood	OL468558	OL468578	OL473526	OL739572	OL473537	[25]
*Cordyceps militaris*	YFCC 6587	Lepidoptera	MN576762	MN576818	MN576988	MN576878	MN576932	[26]
*Cordyceps militaris*	YFCC 5840	Lepidoptera	MN576763	MN576819	MN576989	MN576879	MN576933	[26]
*Cordyceps morakotii*	BCC 55820	Hymenoptera		MF140730	KT261399			[33]
*Cordyceps morakotii*	BCC 68398	Hymenoptera		MF140731	KT261398			[33]
*Cordyceps nabanheensis*	YFCC 8409	Lepidoptera	OL468564	OL468584	OL473532	OL739578	OL473543	[25]
*Cordyceps nabanheensis*	YFCC 8410	Lepidoptera	OL468565	OL468585	OL473533	OL739579	OL473544	[25]
*Cordyceps neopruinosa*	BCC 91361	Lepidoptera		MT003047	MT017858		MT017838	[29]
*Cordyceps neopruinosa*	BCC 91362	Lepidoptera		MT003048	MT017859	MT017818	MT017839	[29]
*Cordyceps nidus*	HUA 186125	Araneae	KC610778	KC610752	KC610722		KC610711	[37]
*Cordyceps nidus*	HUA 186186	Araneae	KY360301	KC610753	KC610723	KY360297		[37]
*Cordyceps ninchukispora*	EGS 38.165	Plant	EF468991	EF468846	EF468795	EF468900		[12]
*Cordyceps ninchukispora*	EGS 38.166	Plant	EF468992	EF468847	EF468794	EF468901		[12]
*Cordyceps ningxiaensis*	HMJAU 25074	Diptera		KF309671				[38]
*Cordyceps ningxiaensis*	HMJAU 25076	Diptera		KF309673				[38]
*Cordyceps nototenuipes*	YFCC 8404	Lepidoptera	OL468559	OL468579	OL473527	OL739573	OL473538	[25]
*Cordyceps nototenuipes*	YFCC 8405	Lepidoptera	OL468560	OL468580	OL473528	OL739574	OL473539	[25]
*Cordyceps oncoperae*	ARSEF 4358	Lepidoptera	AF339581	AF339532	EF468785	EF468891	EF468936	[12,39]
*Cordyceps polyarthra*	MCA 996	Lepidoptera	MF416597	MF416543	MF416487	MF416644		[27]
*Cordyceps polyarthra*	MCA 1009	Lepidoptera	MF416598	MF416544	MF416488	MF416645		[27]
*Cordyceps polystromata*	YFCC 1610885	Lepidoptera	OQ878491	OQ878487	OQ868508	OQ868514	OQ868511	This study
*Cordyceps polystromata*	YFCC 1610886	Lepidoptera	OQ878492	OQ878488	OQ868509	OQ868515	OQ868512	This study
*Cordyceps pruinosa*	ARSEF 5413	Lepidoptera	AY184979	AY184968	DQ522351	DQ522397	DQ522451	[40]
*Cordyceps qingchengensis*	MFLU 17-1022	Lepidoptera	MK761206	MK761211	MK770630			[34]
*Cordyceps rosea*	spat 09-053	Lepidoptera	MF416590	MF416536	MF416480	MF416637	MF416442	[27]
*Cordyceps roseostromata*	ARSEF 4871	Coleoptera	AF339573	AF339523				[39]
*Cordyceps sapaensis*	YFCC 5833	Lepidoptera	MN576764	MN576820	MN576990	MN576880	MN576934	[26]
*Cordyceps sapaensis*	YFCC 1610884	Lepidoptera	OQ878490	OQ878486	OQ868507	OQ868513	OQ868510	This study
*Cordyceps shuifuensis*	YFCC 5230	Lepidoptera	MN576721	MN576777	MN576947	MN576837	MN576893	[26]
*Cordyceps simaoensis*	YFCC 8406	Lepidoptera	OL468561	OL468581	OL473529	OL739575	OL473540	[25]
*Cordyceps simaoensis*	YFCC 8407	Lepidoptera	OL468562	OL468582	OL473530	OL739576	OL473541	[25]
*Cordyceps simaoensis*	YFCC 8408	Lepidoptera	OL468563	OL468583	OL473531	OL739577	OL473542	[25]
*Cordyceps* sp.	CBS 102184	Arachnid	AF339613	AF339564	EF468803	EF468907	EF468948	[12,39]
*Cordyceps* sp.	EFCC 2535	Coleoptera	EF468980	EF468835	EF468772			[12]
*Cordyceps spegazzinii*	ARSF 7850	Diptera		DQ196435				[41]
*Cordyceps subtenuipes*	YFCC 6051	Lepidoptera	MN576719	MN576775	MN576945	MN576835	MN576891	[26]
*Cordyceps subtenuipes*	YFCC 6084	Lepidoptera	MN576720	MN576776	MN576946	MN576836	MN576892	[26]
*Cordyceps succavus*	MFLU 18-1890	Lepidoptera	MK086058	MK086062		MK084616	MK079353	[42]
*Cordyceps tenuipes*	ARSEF 5135	Lepidoptera	MF416612	JF415980	JF416020	JN049896	JF416000	[27,35]
*Cordyceps tenuipes*	YFCC 4266	Lepidoptera	MN576774	MN576830	MN577000	MN576890	MN576944	[26]
*Cordyceps yinjiangensis*	YJ06221	Hymenoptera			MT577003		MT577002	[43]
*Liangia sinensis*	YFCC 3103	Fungi	MN576726	MN576782	MN576952	MN576842	MN576898	[26]
*Liangia sinensis*	YFCC 3104	Fungi	MN576727	MN576783	MN576953	MN576843	MN576899	[26]

Boldface: data generated in this study.

**Figure 1 jof-09-00676-f001:**
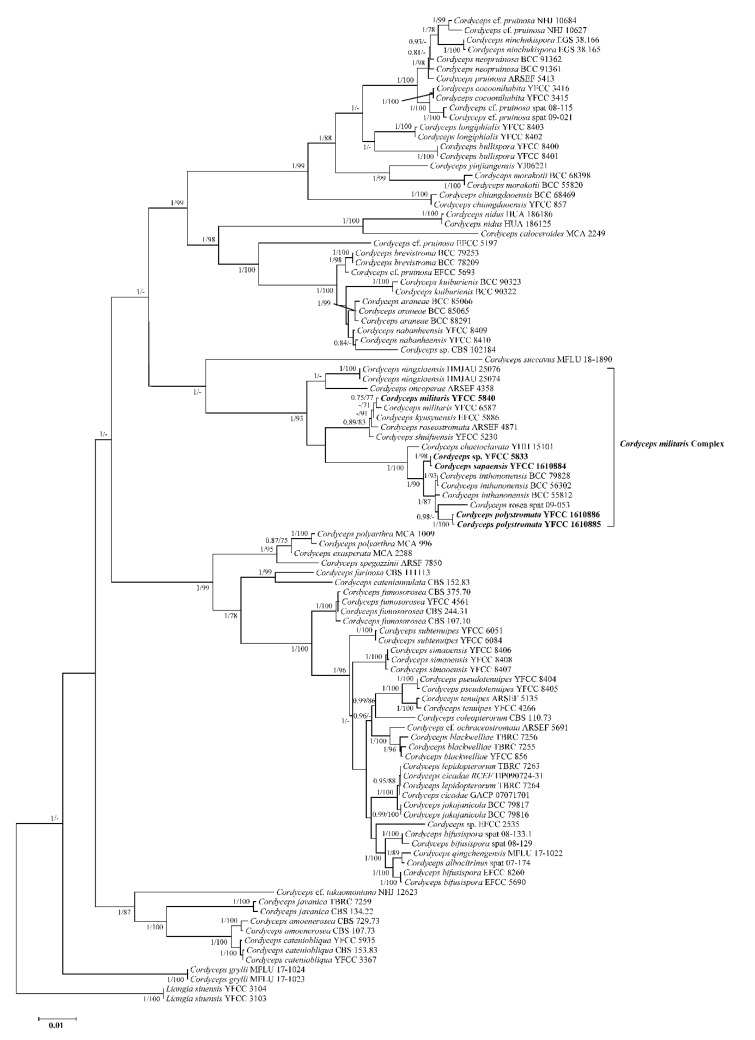
Molecular phylogenetic analyses using the Bayesian Inference (BI) and maximum likelihood (ML) method based on combined nr*SSU*, nr*LSU*, *TEF*, *RPB1*, and *RPB2* sequence data. *Liangia sinensis* YFCC 3103 and *L. sinensis* YFCC 3104 were used as outgroups. Statistical support values (≥0.7/70%) are shown at the nodes for BI posterior probabilities/ML bootstrap support. Isolates in bold type are those analyzed in this study.

### 3.2. Morphological Features

The morphological characteristics of the three *Cordyceps* species as well as photomicrographs of morphological structures are shown in Figure 2, Figure 3 and Figure 4. Detailed fungal morphological descriptions are provided in the Taxonomy section.

### 3.3. Taxonomy

*Cordyceps militaris* (Linnaeus) Fries, Observationes Mycologicae 2: 317 (cancellans) (1818), Figure 2.

MycoBank: MB 237604.

*Clavaria militaris* Linnaeus, Species Plantarum 2: 1182 (1753)

(Basionym)

Sexual morph: Stromata solitary or in groups of 2–3, arising from lepidopteran pupae or larvae buried in soil, cylindrical to clavate, 1.5–9.0 cm long (*n* = 10). Stipes cylindrical, yellowish to orange. Fertile parts clavate, yellowish, reddish-orange, 8–45 × 3.4–6.5 mm (*n* = 10); the perithecial area completely covering the terminal portion of the stroma. Perithecia ovoid, orange to reddish-orange, loosely packed, semi-immersed, 230–556 × 113–319 µm (*n* = 50). Asci cylindrical, 200.0–480.6 × 2.9–4.7 µm (*n* = 50), with a hemispheric apical cap of 3.1–4.5 × 1.8–3.2 µm (*n* = 50). Ascospores filiform, multiseptate, finally breaking into one-celled part-spores, 1.8–4.2 × 0.7–1.6 µm.

Asexual morph: Colonies on PDA fast-growing, 40–45 mm diameter in 14 days at 25 °C, white to yellow, cottony, with protuberant mycelial density at the centrum, reverse yellowish to orange. Hyphae smooth-walled, branched, septate, hyaline, 0.9–2.4 µm wide. Conidiophores smooth-walled, solitary, cylindrical, 3.2–22.5 × 1.4–3.0 µm (*n* = 50). Phialides exist in two types, namely, *Verticillium*- and *Paecilomyces*-phialides. Phialides verticillate on conidiophores, solitary or verticillate on hyphae, *Verticillium*-phialides cylindrical to subulate, 2.8–29.5 × 0.8–3.4 μm (*n* = 50); *Paecilomyces*-phialides swollen or cylindrical at the base tapering to the apex, 5.8–16.5 × 1.4–3.1 μm (*n* = 50). Conidia in chains or heads, hyaline, smooth-walled, one-celled, subglobose to ellipsoidal, 1.8–5.6 × 1.4–3.2 µm (*n* = 100).

Host: Pupa and larva of Lepidoptera.

Habitat: In the soil of evergreen broad-leaf forests, evergreen defoliated broadleaf mixed forests, and coniferous forests.

Distribution: Worldwide.

Material examined: Vietnam, Lao Cai Province, Sa Pa District (22°21′4″ N, 103°46′29″ E, 1931 m above sea level), on pupae and larvae of Lepidoptera buried in forest soil, 30 October 2016, collected by Hong Yu (YHH 933–YHH 944, YHH 5840; living culture: YFCC 933–YFCC 944, YFCC 5840).

Notes: *Cordyceps militaris* is characterized by solitary or several stromata, yellowish to reddish-orange fertile parts, semi-immersed and ovoid perithecia, cylindrical asci, filiform ascospores with multi-septa, short part-spores (https://www.mycobank.org, accessed on 20 April 2023), circular colonies with white to yellow colors, *Verticillium*-like and *Paecilomyces*-like asexual conidiogenous structures, and on the pupae or larvae of Lepidoptera buried in soil [44].

The strain (YFCC 5840) isolated from the pupa of Lepidoptera from Vietnam formed a well-supported clade with a known *C. militaris* isolate (YFCC 6587) (Figure 1). According to microscopic observation, the strain YFCC 5840 showed typical morphological characteristics found in isolates of *C. militaris*. Both the morphological study and phylogenetic analyses supported the isolate YFCC 5840 as being *C. militaris*.

**Figure 2 jof-09-00676-f002:**
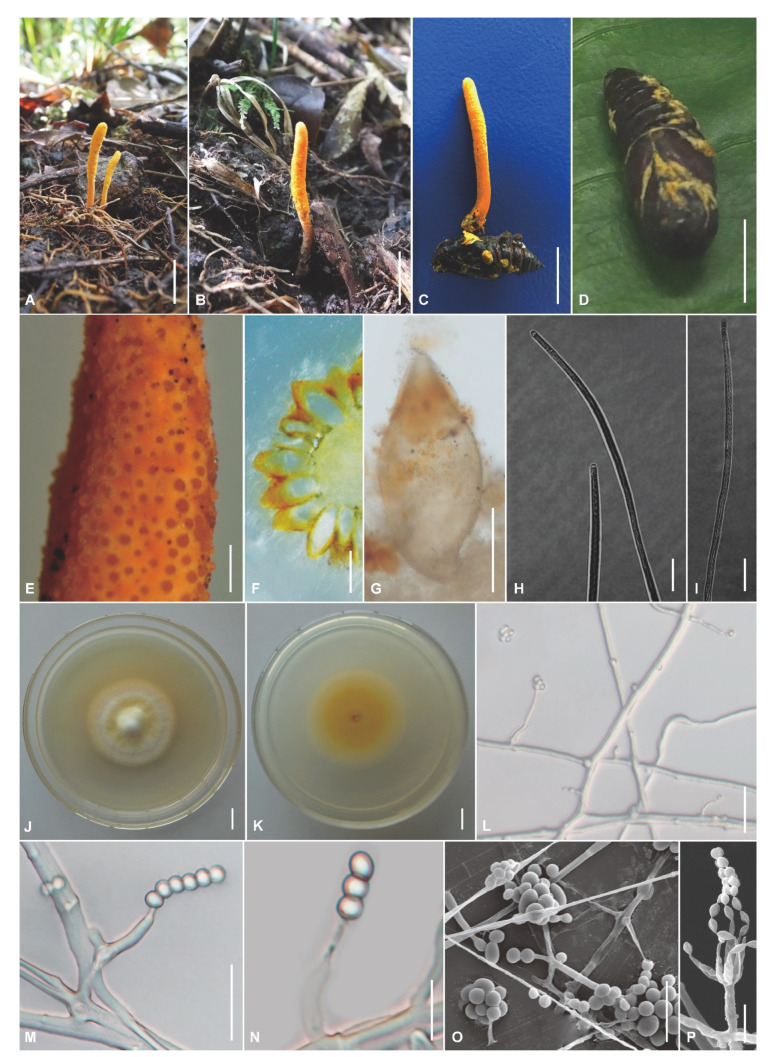
Morphology of *Cordyceps militaris*. (**A**,**B**) Perithecial stromata as encountered in the field; (**C**) Stroma arising from lepidopteran pupa; (**D**) The host of *Cordyceps militaris*; (**E**) Surface of the fertile structure of perithecial stroma showing emerging apical parts of semi-immersed perithecia; (**F**,**G**) Perithecia; (**H**,**I**) Asci; (**J**,**K**) Colony on potato dextrose agar (PDA) medium; (**L**–**P**) Phialides and conidia ((**L**,**O**): *Verticillium*-type; (**M**,**N**,**P**): *Isaria*-type). Scale bars: (**A**,**B**,**D**,**J**,**K**) = 10 mm; (**E**) = 500 μm; (**F**) = 200 μm; (**G**) = 100 μm; (**H**,**I**,**L**) = 20 μm; (**M**,**O**,**P**) = 10 μm; (**N**) = 5 μm.

*Cordyceps polystromata* H. Yu bis, Y. Wang & Q.Y. Dong, sp. nov., Figure 3.

MycoBank: MB 848568.

Etymology: The epithet ‘*polystromata*’ refers to the numerous stromata.

Diagnosis: Differs from *C. militaris* and *C. rosea* in abundant reddish-orange stromata, superficial perithecia, and smaller ovoid to ellipsoidal conidia (1.5–3.7 × 1.2–2.5 µm).

Type: Vietnam, Lao Cai Province, Sa Pa District (22°21′7″ N, 103°46′48″ E, 1931 m above sea level), on a larva of Lepidoptera emerging from the leaf litter on the forest floor, 26 October 2016, Hong Yu (holotype: YHH 1610885; ex-type living culture: YFCC 1610885).

Sexual morph: Stromata arising from the whole body of lepidopteran larvae, gregarious, unbranched, 10–17 mm long. Stipe cylindrical, 4–11 mm long, 1.5–4.2 mm in diameter, orange to reddish-orange, fleshy. The fertile part is cylindrical to clavate, reddish-orange, covered by a spinous surface, 3–14 × 1.6–3.5 mm; the perithecial area fully covers the terminal portion of the stroma. Perithecia superficial, ovoid, reddish-orange, 522–663 × 296–577 µm (*n* = 50). Asci 8-spored, cylindrical, 54.2–172.8 × 4.1–6.5 µm (*n* = 50), with a hemispheric apical cap of 3.5–6.3 × 2.8–4.7 µm (*n* = 50). Ascospores filiform, multiseptate, 51.4–170.5 × 0.9–2.6 µm (*n* = 20), finally breaking into one-celled part-spores, 5.7–7.0 × 1.7–3.2 µm (*n* = 50).

Asexual morph: Colonies on PDA growing fairly well at 25 °C, 33–35 mm in 14 days, white to pale yellow, with high mycelial density, reverse yellowish to orange. Hyphae smooth-walled, branched, septate, hyaline, 2.1–3.7 μm wide. Conidiophores cylindrical, hyaline, smooth-walled, solitary or verticillate, 6.1–43.7 × 1.5–2.9 µm (*n* = 50). Phialides verticillate, in whorls of 2–5, sometimes solitary on hyphae, basal portion cylindrical to narrowly lageniform, tapering gradually or abruptly toward the apex, 6.2–17.2 × 0.9–2.7 μm (*n* = 50). Conidia one-celled, hyaline, smooth-walled, ovoid to ellipsoidal, 1.5–3.7 × 1.2–2.5 µm (*n* = 100). Chlamydospores were not observed.

Other material examined: Vietnam, Lao Cai Province, Sa Pa District, Hoang Lien National Park (22°21′10″ N, 103°46′29″ E, 1989 m above sea level), on a larva of Lepidoptera emerging from the leaf litter on the forest floor, 28 October 2016, Hong Yu (paratype: YHH 1610886; ex-paratype living culture: YFCC 1610886).

Host: Larva of Lepidoptera.

Habitat: The hosts were found in the leaf litter on the forest floor.

Distribution: At present, known only in Sa Pa District, Lao Cai Province, Vietnam.

Notes: *Cordyceps polystromata* is characterized by abundant reddish-orange stromata arising from the whole body of lepidopteran larvae, reddish-orange fertile parts, superficial and ovoid perithecia, cylindrical asci, filiform ascospores with multi-septa, cylindrical part-spores, circular colonies with white to pale yellow colors, and *Paecilomyces*-like asexual conidiogenous structures. It is phylogenetically clustered with *C. chaetoclavata*, *C. inthanonensis*, *C. rosea*, and *C. sapaensis*, but it is distinguished from the four latter species by forming a separate clade in this group (Figure 1). Morphologically, species in this group produce part-spores, except for *C. rosea*, which produces whole ascospores with septations [45]. Among them, only three species, namely, *C. chaetoclavata*, *C. polystromata*, and *C. sapaensis*, have superficial perithecia [26]. Ecologically, *C. inthanonensis*, *C. polystromata*, and *C. rosea* were found to occur on lepidopteran larvae, while *C. chaetoclavata* and *C. sapaensis* were found on lepidopteran pupae buried in soil [26,29,45].

Macro-morphologically, *C*. *polystromata* is very similar to *C. inthanonensis* [29]. They have the same abundant orange to reddish-orange stromata, cylindrical to clavate fertile parts, and host of lepidopteran larvae. However, morphological observation reveals a significant difference in conidia sizes between *C*. *polystromata* (1.5–3.7 × 1.2–2.5 µm) and *C. inthanonensis* (4–7(9) × 1.5–2 μm). *Cordyceps polystromata* can also be distinguished from *C. inthanonensis* by shorter cylindrical asci (54.2–172.8 × 4.1–6.5 µm).

**Figure 3 jof-09-00676-f003:**
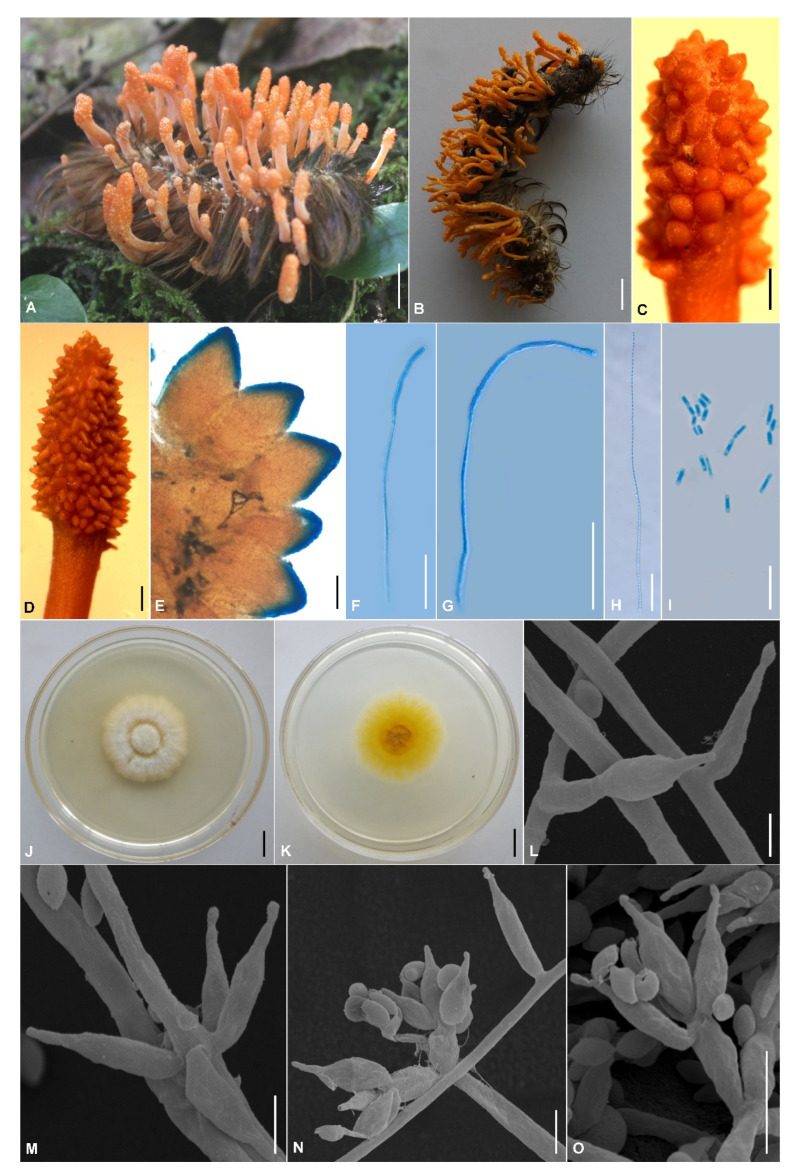
Morphology of *Cordyceps polystromata*. (**A**,**B**) Stromata arising from the host of lepidopteran larvae; (**C**,**D**) Fertile part; (**E**) Perithecia; (**F**,**G**) Asci; (**H**) Ascospores; (**I**) Part-spores; (**J**,**K**) Colony on potato dextrose agar (PDA) medium; (**L**–**O**) Conidiophores, phialides, and conidia. Scale bars: (**A**,**B**,**J**,**K**) = 10 mm; (**C**,**D**) = 500 μm; (**E**) = 200 μm; (**F**–**H**) = 50 μm; (**I**) = 10 μm; (**L**–**O**) = 5 μm.

*Cordyceps sapaensis* H. Yu bis, Y. Wang & Q.Y. Dong, sp. nov., Figure 4.

MycoBank: MB 848569.

Etymology: Named after the location of Sa Pa District where the species was collected.

Diagnosis: *Cordyceps sapaensis* can be distinguished by clavate stromata with banana-shaped fertile parts, large superficial perithecia (587–743 × 341–396 µm), cylindrical phialides (conidiogenous cells), and large ellipsoidal to cylindrical conidia (2.6–7.4 × 1.1–3.1 µm).

Type: Vietnam, Lao Cai Province, Sa Pa District, Hoang Lien National Park (22°19′30″ N, 103°46′50″ E, 2178 m above sea level), on a pupa of Lepidoptera buried in soil, 28 October 2016, Hong Yu (holotype: YHH 1610884; ex-type living culture: YFCC 1610884).

Sexual morph: Stromata arising from pupae of Lepidoptera buried in soil, solitary, unbranched, 18–35 mm long. The stipe is cylindrical, 15–21 mm long, 2.8–4.3 mm in diameter, yellowish to orange, and fleshy. The fertile part is banana-shaped, yellowish, reddish-orange, 8–15 × 3.3–4.8 mm; the perithecial area completely covers the terminal portion of the stroma. Perithecia superficial, crowded, ovoid, reddish-orange, 587–743 × 341–396 µm (*n* = 50). Asci 8-spored, cylindrical, 237.2–472.6 × 3.1–5.4 µm (*n* = 50), with a hemispheric apical cap of 2.5–4.3 × 1.6–2.8 µm (*n* = 50). Ascospores filiform, multiseptate, 230.2–457.8 × 1.6–2.1 µm (*n* = 20), finally breaking into one-celled part-spores, 2.0–4.8 × 1.3–2.2 µm (*n* = 50).

Asexual morph: Colonies on PDA growing fairly well, attaining a diameter of 34–36 mm after 14 days at 25 °C, white to pale yellow, with high mycelial density, reverse cream to yellow. Hyphae hyaline, branched, smooth-walled, 0.9–3.1 µm wide. Phialides arising from aerial hyphae, solitary, sometimes in whorls of 2–5, basal portion cylindrical, tapering gradually toward the apex; 5.1–28.5 µm long, 1.5–3.1 µm wide at the base, and 1.3–2.1 µm wide at the apex (*n* = 50). Conidia one-celled, hyaline, smooth-walled, ellipsoidal to cylindrical, 2.6–7.4 × 1.1–3.1 µm (*n* = 100). Chlamydospores present, one-celled, solitary, eggplant-shaped or oval to pyriform, hyaline becoming brown, thick, and smooth-walled.

Other material examined: Vietnam, Lao Cai Province, Sa Pa District (22°21′6″ N, 103°46′41″ E, 1948 m above sea level), on a pupa of Lepidoptera buried in soil, 26 October 2016, Hong Yu (paratype: YHH 5833; ex-paratype living culture: YFCC 5833).

Host: Pupa of Lepidoptera.

Habitat: In the soil of evergreen broad-leaf forests.

Distribution: At present, known only in Sa Pa District, Lao Cai Province, Vietnam.

Notes: Regarding phylogenetic relationships, *C. sapaensis* forms a distinct lineage in the *C. militaris* complex, and it is closely related to *C. chaetoclavata*, *C. inthanonensis*, *C. polystromata*, and *C. rosea* (Figure 1). Morphologically, *C. sapaensis* is similar to *C. chaetoclavata* and *C. rosea* by sharing single, unbranched, fleshy, and cylindrical stipes, and yellowish to reddish-orange stromata. According to the original description of *C. chaetoclavata*, it has spinous fertile parts and superficial lageniform perithecia (402–610 × 280–427 µm) [26]. However, *C. sapaensis* differs from *C. chaetoclavata* by its banana-shaped fertile parts and longer superficial perithecia with an ovoid shape (587–743 × 341–396 µm). Additionally, our morphological observation reveals a significant difference between *C. rosea* and *C. sapaensis*. *Cordyceps rosea* has rose stromata (11 mm long), immersed perithecia, and the host of lepidopteran larvae [45], whereas *C. sapaensis* has longer stromata (18–35 mm long) with yellowish to orange colors, superficial perithecia, and the host of lepidopteran pupae.

**Figure 4 jof-09-00676-f004:**
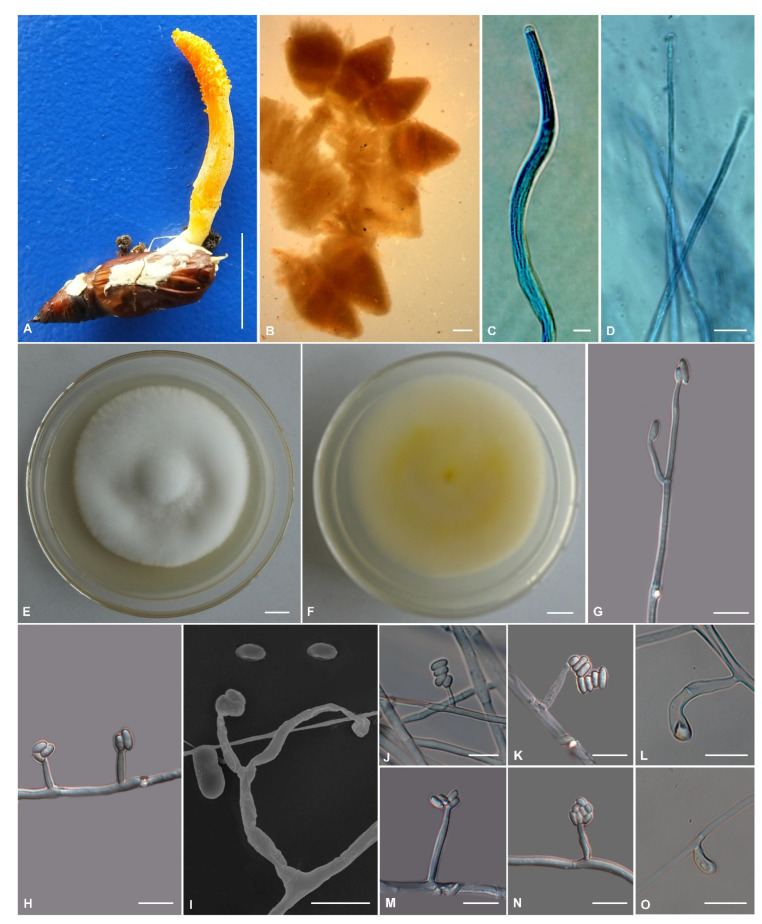
Morphology of *Cordyceps sapaensis*. (**A**) Fungus on the pupa of Lepidoptera; (**B**) Perithecia; (**C**,**D**) Asci; (**E**,**F**) Colony on potato dextrose agar (PDA) medium; (**G**–**K**,**M**,**N**) Phialides and conidia; (**L**,**O**) Chlamydospores. Scale bars: (**A**,**E**,**F**) = 10 mm; (**B**) = 200 µm; (**D**,**G**–**O**) = 10 µm; (**C**) = 5 µm.

## 4. Discussion

The phylogenetic and morphological analyses presented here strongly supported the hypothesis that the fungal materials collected in Vietnam belonged to *C. militaris* and two hidden species in the *C. militaris* complex. The molecular phylogeny showed well-supported clades for *Cordyceps*, thereby supporting the descriptions of *C. polystromata* (Figure 3) and *C. sapaensis* (Figure 4) as new taxa, as well as *C. militaris* (Figure 2) as a known species.

In the *C. militaris* complex, the macromorphology of numerous species is similar, which could easily lead to misidentification. *Cordyceps chaetoclavata*, *C. militaris*, *C. ningxiaensis*, *C. oncoperae*, *C. rosea*, *C. roseostromata*, *C. sapaensis*, and *C. shuifuensis* shared numerous similar morphological characteristics of sexual morphs, viz., single, fleshy, and cylindrical stipes; yellowish, orange to reddish stromata; cylindrical asci with thickened ascus apex; and filiform and multiseptate ascospores [26,38,45,46,47,48]. *Cordyceps inthanonensis* and *C. polystromata* had the same abundant orange to reddish-orange stromata arising from the whole body of lepidopteran larvae, cylindrical to clavate fertile parts, ovoid perithecia, cylindrical asci, filiform ascospores with multi-septa, and cylindrical part-spores [29]. Species in the *C. militaris* complex were found to occur on lepidopteran pupae or larvae except for *C. ningxiaensis* (fly pupae) and *C. roseostromata* (larvae of Coleoptera) [26,29,38,45,46,47,48]. Both the macroscopic and microscopic observations conducted throughout the investigation revealed the extensive overlap in morphological characters and the lack of distinctive phenotypic variation, thus supporting the notion of cryptic species in a species complex.

In the current study, a comprehensive morphological and phylogenetic investigation of the *C. militaris* complex in Vietnam was conducted. Both the macroscopic and microscopic observation of the collections compared with other known species in the *C. militaris* complex revealed some obvious differences, although the morphological features overlapped generally, thereby supporting the notion of cryptic species in a species complex (Table 2). The species described in this study were all distinct from other closely related species of *Cordyceps*; *C. militaris* had stromata that were usually single or sometimes in groups of 2–3 stromata, semi-immersed and ovoid perithecia, short part-spores, *Verticillium*-like and *Paecilomyces*-like asexual conidiogenous structures, and they were found on the pupae or larvae of Lepidoptera buried in soil; *C. polystromata* produced abundant orange to reddish-orange stromata arising from the whole body of lepidopteran larvae, cylindrical to clavate fertile parts with a spinous surface, superficial ovoid perithecia, and *Paecilomyces*-like asexual conidiogenous structures; and *C. sapaensis* had a single yellowish to orange stroma, banana-shaped fertile parts, and longer superficial perithecia with ovoid shape. Molecular phylogenetic analyses based on the combined dataset of nr*SSU*, nr*LSU*, *TEF*, *RPB1*, and *RPB2* also supported the existence of the known species and two distinct species in the *C. militaris* complex (Figure 1), thus emphasizing the importance of morphological and molecular identification.

## Figures and Tables

**Table 2 jof-09-00676-t002:** Morphological comparison of species in the *C. militaris* complex.

Species	Stromata (mm)	Fertile Parts (mm)	Perithecia (μm)	Asci (μm)	Ascospores (μm)	Part-Spores (μm)	Phialides (μm)	Conidia (μm)	References
*Cordyceps chaetoclavata*	Solitary, 23 × 0.8	Clavate, covered by a spinous surface, 5.6 × 0.7–1.1	Superficial, 402–610 × 280–427	Cylindrical, 274–385 × 3.7–4.8	Filiform, multiseptate, 127–260 × 0.9–1.2	Cylindrical, 3–12 μm long			[26]
*Cordyceps inthanonensis*	Multiple, 6–25 mm long	Cylindrical to clavate, 3–5 mm wide	Semi-immersed, ovoid, 600–720 × 220–420	Cylindrical, 450–600 × 4–6	Filiform, multiseptate, 400–550 μm long	Cylindrical, 3–4 × 1–1.5	Solitary, cylindrical at the base tapering to the apex, (12)14–18.5(20) × 1.5–3	Cylindrical, 4–7(9) × 1.5–2	[29]
*Cordyceps kyusyuensis*	Multiple, 15–20 mm long	Cylindrical, 10–12 mm long	Semi-superficial, ovoid, 410–580 × 210–330	4 μm wide		4–5 × 1		Ovoid, 2 × 1.5	[46]
*Cordyceps militaris*	Solitary or in groups of 2–3, 15–90 mm long	Clavate, 8–45 × 3.4–6.5	Semi-immersed, ovoid, 230–556 × 113–319 µm	Cylindrical, 200.0–480.6 × 2.9–4.7	Filiform, multiseptate	1.8–4.2 × 0.7–1.6	Solitary or verticillate, *Verticillium*-type: 2.8–29.5 × 0.8–3.4, *Paecilomyces*-type: 5.8–16.5 × 1.4–3.1	Subglobose to ellipsoidal, 1.8–5.6 × 1.4–3.2	This study
*Cordyceps ningxiaensis*	1 to 2 in a group	Spherical to ovoid, 1.2–3 × 1.2–2.8	Immersed, ellipsoid to ovoid, 288–400 × 103–240	Cylindrical, 168–205 × (3.7–)4.1–5.5(–6.6)	Filiform, irregularly multiseptate	3.6–7.8 × 1.0–1.4			[38]
*Cordyceps oncoperae*	Solitary to multiple, up to 35 mm long	Clavate, usually with acute apices, 4–10 × 2–3	Ovoid, 350–410 × 180–230(–380)	Cylindrical, (168–)200–224(–256) × (5–)6–6.5	Filiform, multiseptate, 104–139 × 1.5–2				[48]
*Cordyceps polystromata*	Multiple, 10–17 mm long	Cylindrical to clavate, covered by a spinous surface, 3–14 × 1.6–3.5	Superficial, ovoid, 522–663 × 296–577	Cylindrical, 54.2–172.8 × 4.1–6.5	Filiform, multiseptate, 51.4–170.5 × 0.9–2.6	5.7–7.0 × 1.7–3.2	Verticillate, 6.2–17.2 × 0.9–2.7	Ovoid to ellipsoidal, 1.5–3.7 × 1.2–2.5	This study
*Cordyceps rosea*	Solitary, 11 mm long	Clavate	Immersed, ovoid, 330–380 × 160–230	100 × 3–4	Filiform, multiseptate, 120 × 1–1.5			Navicular, 4–5 × 1	[45]
*Cordyceps roseostromata*	Solitary to multiple	Subglobose to clavate, 1.2–5 × 1.5–2.2	Superficial, pyriform, 280–300 × 140–160	3–3.5 μm wide		4–5 × 1			[47]
*Cordyceps sapaensis*	Solitary, 18–35 mm long	Banana-shaped, 8–15 × 3.3–4.8	Superficial, crowded, ovoid, 587–743 × 341–396	Cylindrical, 237.2–472.6 × 3.1–5.4	Filiform, multiseptate, 230.2–457.8 × 1.6–2.1	2.0–4.8 × 1.3–2.2	Solitary or verticillate, cylindrical, 5.1–28.5 µm long, 1.5–3.1 µm wide at the base, 1.3–2.1 µm wide at the apex	Ellipsoidal to cylindrical, 2.6–7.4 × 1.1–3.1	This study
*Cordyceps shuifuensis*	Solitary, 25 mm long	Clavate, 4 × 1.5	Pseudoimmersed, ovoid, 450–620 × 300–430	Cylindrical, 275–510 × 3.5–5.2	Filiform, multiseptate, 180–410 × 1.2–1.7	Cylindrical, 2.8–6.5 μm long	Solitary or verticillate, cylindrical or subulate, 4.7–20 um long, 1.1–2.0 μm wide at the base, 0.4–2.1 μm wide at the apex	Macroconidia clavate to oblong-ovate, 5.1–11.8 × 1.3–2.4; microconidia globose to ellipsoidal, 1.8–3.0 × 1.6–2.5	[26]

Boldface: data generated in this study.

## Data Availability

The DNA sequences data obtained in this study have been deposited in GenBank. The accession numbers can be found in the article (Table 1).

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
