# Peer review of "Molecular Phylogeny and Morphology Reveal Cryptic Species in the Cordyceps militaris Complex from Vietnam"

_jof, 2023, doi:10.3390/jof9060676_

Round 1

Reviewer 1 Report

SUGGESTIONS TO AUTHORS:

The manuscript is a very interesting contribution to the Cordyceps taxonomy, specially of the C.  militaris complex. The text is well written and needs no alteration.

My suggestions are:

- In the Keywords: change “morphology” to other word, since this word is already found in the title;

- line 140 to 144: include the authors of each species (or in the list in Table 1).

- line 145: the name Table 1 appears two times, being the second better assigned as “Species list” or something similar.

- line 145: it is important to include authors of each species in the list.

- Table 1: in some sentences of the table adjust the “parenthesis”, since sometimes the same are alone on the next line as in the example below (review all the table):

- line 159: “Mycologicae” instead of “mycologicae”.

- Figure 2: the images (B) and (C) have the same scale bar but (C) is closer than (B). Adjust the scale and cut off the ruler include a normal scale bar in (C) image;

- Figure 3: the first image (A) is a close up and the second image (B) is a distant photograph but the scale is the same (10 mm) for both. In the (B) image the scale bare needs to be reduced to represent 10 mm.

- line 220: compare “growing fairly well at 25°C, 33–35 mm in 14 days” with line 273 “moderately fast-growing, attaining a diameter of 34–36 mm 274 after 14 days at 25°C” and line 171 “fast-growing, 40–45 mm diameter”. The diameters are overlapped in the new species, so use the same “growing fairly well” for both diameters (for the new species).

Reviewer 2 Report

The manuscript entitled " Molecular phylogeny and morphology reveal cryptic species in the Cordyceps militaris complex from Vietnam” is appropriate for the journal. It is an original and relevant contribution to generate knowledge about the new species of entomopathogenic fungi, located in tropical regions that has an extremely rich biodiversity.

Very good work for the authors, congratulations.

The article can be accepted for publication, attending to the comments of the manuscript.

Some specific comments:

L46- RMB, describe the acronyms

L104 - All primer names and their sequences must be specified (Please add a Table in the manuscript or as supplementary material). Primers from reference 19 do not show the same name as indicated in the manuscript.

In Table 1, the first column title indicates “Table 1.”, instead of “strain” or “specimen”.

Figure 1 image quality is low, perhaps it is the PDF generator from the Journal and not the image file itself. Please use a higher-quality image if possible.

In Fig 1, why is Cordyceps sp. YFCC 5833 shown in bold font? It is confusing because it may indicate it was isolated by this work but it was previously reported by [32].

The authors associate the fungal isolates as arthropod-pathogenic or capable of infecting insects; however, there is no strong evidence of this, for example a bioassay to confirm pathogenicity (Koch's postulates), under controlled conditions, I consider it should be a basic element to describe new species of this nature.
